# Comparison of Computational Intelligence Methods Based on Fuzzy Sets and Game Theory in the Synthesis of Safe Ship Control Based on Information from a Radar ARPA System

**Józef Lisowski ***[image ID] **and Mostefa Mohamed-Seghir**

Department of Ship Automation, Gdynia Maritime University, 81-225 Gdynia, Poland;
m.mohamed-seghir@we.umg.edu.pl
* Correspondence: j.lisowski@we.umg.edu.pl; Tel.: +48-694-458-333

**Abstract:** This article presents safe ship control optimization design for navigator advisory system. Optimal safe ship control is presented as multistage decision-making in a fuzzy environment and as multistep decision-making in a game environment. The navigator's subjective and the maneuvering parameters are taken under consideration in the model process. A computer simulation of fuzzy neural anticollision (FNAC) and matrix game anticollision (MGAC) algorithms was carried out on MATLAB software on an example of the real navigational situation of passing three encountered ships in the Skagerrak Strait, in good and restricted visibility at sea. The developed solution can be applied in decision-support systems on board a ship.

**Keywords:** radar; fuzzy sets theory; artificial neural network; game theory; safe ship trajectory; computer simulation; computer decision support

---

## 1. Introduction

One of the most important transport issues is the safe control of movement, measured by the probability of collision risk when passing ships on the route, using information from the radar anticollision system [1,2]. In practice, there are many safe trajectories for the ship, from which an optimal trajectory can be chosen that ensures minimum collision risk and the smallest path loss on passing encountered ships [3–5]. The development of modern information technologies creates appropriate opportunities for the automation of navigation and construction of decision support systems to safety control the movement of a ship (Figure 1).

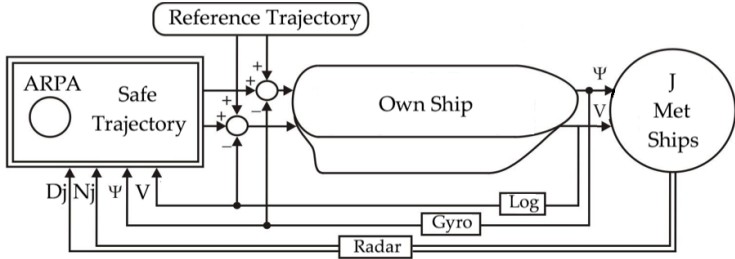

**Figure 1.** Safe ship-control system: $D_j$—distance to j-th met ship, $N_j$—bearing to j-th met ship, $\psi$—course of the own ship, V—speed of the own ship, J—number of met ships.

According to Lloyd's statistics, in about 87% of marine accidents, the cause of ship collisions is the navigator's subjectivity in maneuvering decisions, often under conditions of ambiguity and conflict.

Therefore, among the many possibilities of describing this process, models of fuzzy control and game control become useful [6–8].

The methods of static and dynamic optimization used so far, evolutionary algorithms or particle swarm methods, do not include the fuzzy and game properties of the real anticollision problems of the ship [9,10].

Therefore, the aim of this paper is to determine an optimal and safe ship trajectory using the theory of fuzzy sets and game theory (Figures 2 and 3).

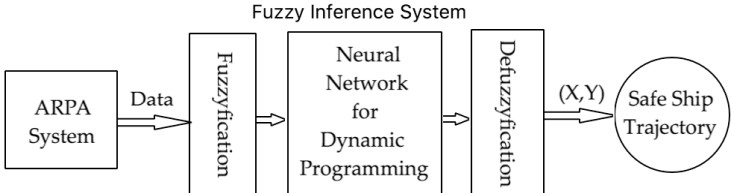

**Figure 2.** Fuzzy control system.

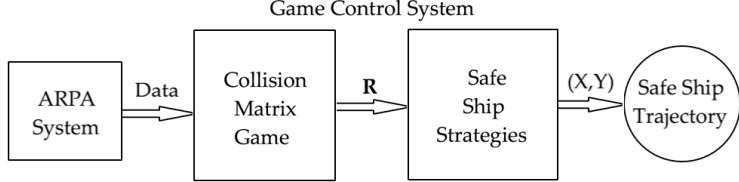

**Figure 3.** Game control system.

## 2. Kinematic Model of the Ship

In practice, the kinematics parameters of passing ships at sea in the form of:

- speed $V_j$,
- course $\psi_j$,
- distance of the closest point of approach $DCPA_j = D^j_{min}$,
- time to the closest point of approach $TCPA_j = T^j_{min}$.

These are identified by the automatic radar plotting aids (ARPA) anticollision system (Figure 4).

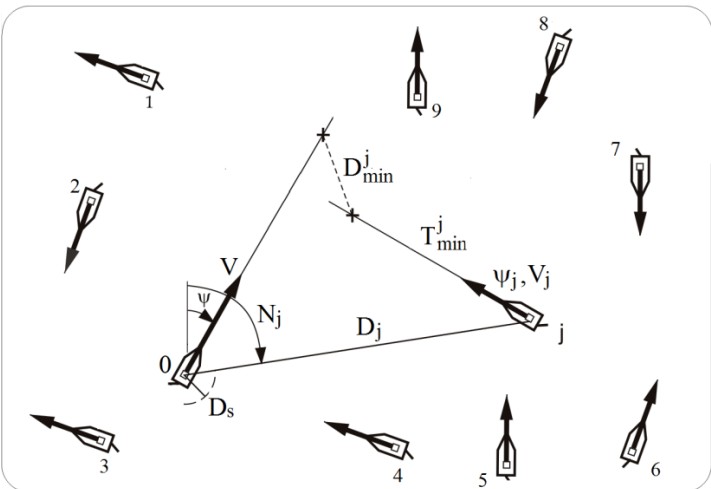

**Figure 4.** Situation of passing own ship with other ships.

The kinematic model of the control can be presented in the form of state equation

$$X_{k+1} = f(X_k, U_k), k = 1, 2, \dots, n \tag{1}$$

where: $(X_{k+1}, X_k)$ is the set of real ship position co-ordinates, and $U_k$ is the control set [11].

The process ends when the ship reaches the back points called the final states

$$W = \left\{ a_{p+1}, a_{p+2}, a_n \right\} \tag{2}$$

The set of final states must meet this condition

$$\left. \begin{matrix} \psi_{opt} = \psi_S \\ V_{opt} = V_S \\ \mu_R \leq \mu_{Rsafe} \end{matrix} \right\} \tag{3}$$

where: $(\psi_{opt}, V_{opt})$ are the optimal own-ship course and speed, $(\psi_S, V_S)$ are set point own-ship course and speed, and $\mu_R$ is the membership function of collision risk [12].

## 3. Fuzzy Control Model of the Process

### 3.1. Membership Function of Fuzzy Goal

Different security assessments made by navigators can be described as the membership function of the fuzzy goal, allowing a subjective assessment of

$$\mu_G(k, j) = 1 - \frac{1}{\exp(\lambda_d(k, j)DCPA_j^2)} \tag{4}$$

where $\lambda_d$ is the navigator's subjective parameter.

### 3.2. Membership Function of Fuzzy Constraints

The membership function of the fuzzy constraints can be defined as

$$\mu_C(k) = \frac{1}{\exp(\lambda_c(k)(V \cos \psi(k) - V \cos \psi(k-1))t_k^2)} \tag{5}$$

where $\lambda_c$ is the navigator's subjective parameter.

### 3.3. Membership Function of Fuzzy Collision Risk

Ships that can take part in a collision should be sorted according to the degree of threat by the collision-risk rating index. In many works, the ship's domain is treated as an assessment of collision risk [8]. In this article, the collision-risk transfer function was used as collision-risk assessment

$$\mu_R(k, j) = \frac{1}{\exp(\lambda_{rd}(k, j)DCPA_j^2 + \lambda_{rt}(k, j)TCPA_j^2)} \tag{6}$$

where $\lambda_{rd}$ and $\lambda_{rd}$ are the navigator's subjective parameters.

The fuzzy-set decision is determined as result of an operation of the fuzzy set of a goal and fuzzy set of constraints

$$\mu_D(., .) = \mu_C(., .) * \mu_G(., .) \tag{7}$$

### 3.4. Fuzzy Neural Anticollision (FNAC) Algorithm

To present the idea of dynamic programming in this case, we first have to present the task in a slightly fuller form

$$
\begin{aligned}
\mu_D(u_0, ..., u_{N-1}|X_0) = \\
\max_{u_0, ..., u_{N-1}} [\mu_C^0(u_0) \wedge \mu_G^1(X_1) \wedge \mu_C^1(u_1) \wedge \mu_G^2(X_2) \wedge ... \\
... \wedge \mu_C^{N-1}(u_{N-1}) \wedge \mu_G^N(f(X_{N-1}, u_{N-1}))]
\end{aligned}
\tag{8}
$$

After transformation and maximization in relation to controls $(u_0, u_1, \ldots, u_{N-1})$, we obtain the following system of recursive equations

$$
\begin{cases}
\mu_G^{N-i}(X_{N-i}) = \max_{u_{N-i}}[\mu_C^{N-i}(u_{N-i}) \wedge \mu_G^{N-i+1}(f(X_{N-i+1}))] \\
X_{N-i+1} = f(X_{N-i}, u_{N-i}) \; i = 0, 1, ..., N
\end{cases}
\tag{9}
$$

Referring to Formula (9), going back from stage $t = N$ to $t = 0$, at each stage there are two phases: minimization and maximization. Such operations can be implemented using the special neural network proposed in [13]. The traditional artificial neural network does not perform the minimum and maximum operations of a finite set. Appropriate neurons were proposed by Rocha, which will be presented in the next sections [14–18].

### 3.4.1. Neural Network

The operations presented above require special neurons that can be used in an artificial neural network. Such neurons were proposed by Rocha [19], neurons of maximum type and minimum type. We assume that the neuron has n inputs $(b_1, b_2 \ldots, b_n)$ and the weighted sum of these n inputs is defined by the following formula

$$
y = \sum_{k=1}^{n} w_k b_k
\tag{10}
$$

where $w_k$ are the synapse weights connecting the input neurons.

The following pattern shows that the resulting obtained value u is encoded as the axonal activation $b_p$ of the postsynaptic neuron.

$$
b_p = \begin{cases}
1 & \text{if} \quad u \geq \alpha_2 \\
f(u) & \text{if} \quad \alpha_1 \leq u \leq \alpha_2 \\
0 & \text{otherwise}
\end{cases}
\tag{11}
$$

We have introduced two axonal thresholds $\alpha_1$, $\alpha_2$, which are defined by polarized neurons, where f transition function. We can now define the maximum-type neuron and the minimum-type neuron.

Maximum-Type Neuron

Defined as such whose axonal threshold $\alpha_t$ in stage t is described as

$$
\alpha(t) = \begin{cases}
1 & \text{if} \quad t = 0 \\
b_p(t-1) & \text{otherwise}
\end{cases}
\tag{12}
$$

However, axionic activation $b_p$ is presented as

$$
b_p(t) = \begin{cases}
\alpha(t) & \text{if } u(t) \leq \alpha(t) \\
u(t) & \text{otherwise}
\end{cases}
\tag{13}
$$

where u(t) is the postsynaptic activation at stage t [20].

Combining the two functions, at the output of the max neuron, we obtain a maximum value of the inputs if weight $w_k = 1$

$$b_p(t) = \max_{k=1,2,...,t} [w_k b_k] \qquad (14)$$

Minimum-Type Neuron

Defined as a neuron whose axionic threshold $\alpha_t$ at stage t is

$$\alpha(t) = \begin{cases} 1 & \text{if } t = 0 \\ b_p(t-1) & \text{otherwise} \end{cases} \qquad (15)$$

However, axionic activation

$$b_p(t) = \begin{cases} \alpha(t) & \text{if } u(t) \geq \alpha(t) \\ u(t) & \text{otherwise} \end{cases} \qquad (16)$$

The equations show that the output of the min neuron encodes at least the minimum value of the inputs if weight $w_k = 1$

$$b_p(t) = \min_{k=1,2,...,t} [w_k b_k] \qquad (17)$$

In this way, defined neurons allow to build the neural network to solve the task of the optimal safe ship trajectory.

### 3.4.2. Structure of Neural Networks in Relation to Multistage Control

The neural network proposed in Reference [13] allows solving the task presented above. Its structure consists of alternating layers of minimum and maximum neurons. Weights values of neuron entrances are not given by learning in the ordinary sense, but result from the description the task, i.e., state transitions, fuzzy constraints, and fuzzy goals. Therefore, in order for the structure of the neural network to properly work, it is necessary to determine the connections between minimum and maximum neurons on the same layer, the maximum neurons of the preceding layer, and the minimum neurons on a current layer.

In the following, the neurons are described as

- $M^i_k$—max neuron at stage k,
- $m^i_k$—min neuron at stage k.

### 3.4.3. Generating Interconnections between Max and Min Neurons at the Same Layer

The connection of the two types of neurons from the same layer is done using the state-transitions equations, the connection function is as

$$W(m^i_k, M^j_k) = \begin{cases} 1 & \text{if } f_{N-k}(q_R(m^i_k), q_T(M^j_k)) \neq 0 \\ 0 & \text{otherwise} \end{cases} \qquad (18)$$

where $q_R(m^i_k)$ is the number of receptor control, $q_T(M^i_k)$ is the number of relay states, $f_{N-k}(.,.)$ is the state-transition equation. A value of 1 means there is a connection, while 0 means no connection.

### 3.4.4. Generating Interconnections between Max Neurons and Min Neurons at the Given Layer

The combination of max neuron $M^j_{k-1}$ (layer k − 1) and min neuron $m^i_k$ (layer k) is executed by the number of receptors $q_R(m^i_k)$ and the number of the relay. It allows to obtain state $x_{N-k}$ using the state equation, running neuron driver $q_C(m^i_k)$. That can be presented as the equation

$$q_C(m^i_k) = x_{N-k} = f_{N-k}(q_R(m^i_k), q_T(M^j_k))$$ (19)

This driver is designed to send to all max neurons in layer (k − 1) the number of receptors $q_C(m^i_k)$. Neurons, which, due to the sent value, are activated, have the same value as receptor $q_R(M^l_{k-1})$. Just like in computer networks, between neuron $m^i_k$ and $M^l_{k-1}$, a connection is established, which can be defined as

$$W(M^l_{k-1}, m^j_k) = \begin{cases} 1 & \text{if } q_R(M^l_{k-1}) = q_T(m^i_k) = q_C(m^i_k) \\ 0 & \text{otherwise} \end{cases}$$ (20)

The connections presented in this fashion give the possibility to form an algorithm based on an artificial neural network, which will emulate solving the problem of dynamic programming in fuzzy environment, that is, the fuzzy neural anticollision (FNAC) algorithm [21].

The structure of the neural network to determine the safe ship trajectory is atypical, the network consists of six stages, in the first stage there are two layers of neurons, one max neuron and nine min neurons. The next has 9 max neurons and 32 min neurons. However, in third stage there are 25 max neurons and 38 min neurons. The penultimate stage has nine max neurons and nine min neurons. The last has one max neurons. The neurons weight result from the function state transitions and the membership function of fuzzy constraints and the membership function fuzzy goals.

Output step is to find a series of connections of maximum neurons, whose outputs have the highest value fuzzy decision $\mu_{Dmax}$.

The initialization step it is to create a neural network. As in the dynamic programming steps proceed from the latter to zero (initial).

Thus, the steps of numbering 0 (the last stage with respect to $X_0$) to N (last step k, the first with respect to $X_0$) initialize the first layer of neurons maximum at stage k = 0, and increase by 1 (k = 1). From this step on, in each subsequent step we first initialize the layer of minimum neurons and then the layer of maximum neurons (Figure 5).

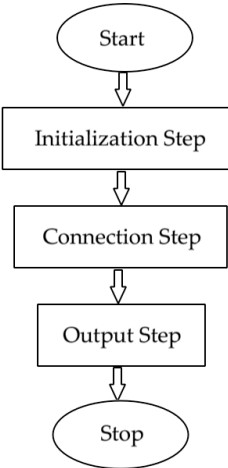

**Figure 5.** Fuzzy neural anticollision (FNAC) algorithm block diagram.

The connection step, as the name suggests, is to combine the maximum and minimum neurons of the same stage k (Phase 1), the maximum neurons at stage k − 1, and the minimum neurons at stage k (Phase 2). The output step is to find a series of connections of maximum neurons whose outputs have the highest value, $\mu_{Dmax}$.

Of course, this is also true for control, which made it possible to obtain a value $\mu_D(u_0{}^*, \dots, u_{N-1}{}^* | X_0)$ so there is also a connection with the minimum neurons. This time, these values are being sought in an order according to the initial state, that is $x_0$ to $x_N$ (final state).

The initial state and the final state are single (this is related to the maneuvers of the ship). In some structures of the neural network, this may occur more than once, in the initial and final states, and, in this case, the connection may vary depending on the selection state at the initial stage.

Using the fuzzy toolbox and neural toolbox contained in the MATLAB_R2016a software, the fuzzy neural anti-collision (FNAC) computer program was designed for the determination of the safe own-ship trajectory in a collision situation [11].

## 4. Game Control Model of the Process

### 4.1. Base-Differential Game Model

The most general description of the own ship passing j other encountered ships is the model of a differential game of moving control objects (Figure 6).

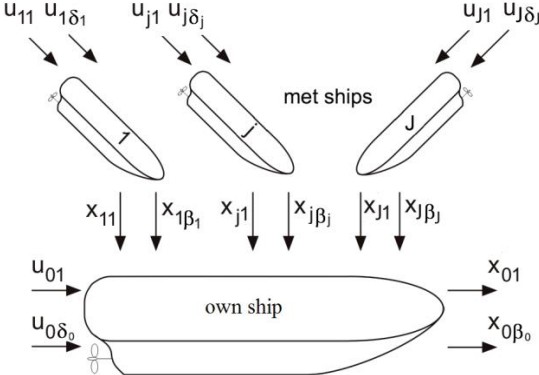

**Figure 6.** Block diagram of the basic model of differential game.

The properties of the process are described by the state equation

$$
\begin{aligned}
\dot{x}_i = f_i \big[ &(x_{0,\beta_0}, x_{1,\beta_1}, \dots, \ x_{j,\beta_j}, \dots, \ x_{J,\beta_J}), \\
&(u_{0,\delta_0}, u_{1,\delta_1}, \dots, \ u_{j,\delta_j}, \dots, \ u_{J,\delta_J}), \ t \big] \\
&j = 1, \ 2, \dots, \ J
\end{aligned}
\tag{21}
$$

where $\vec{x}_{0,\beta_0}(t)$ is the $\beta_0$ dimensional vector of the process state of the own-ship determined in a time span $t \in [t_0, t_k]$; $\vec{x}_{j,\beta_j}(t)$ is the $\beta_j$ dimensional vector of the process state for the j-th met ship; $\vec{u}_{0,\delta_0}(t)$ is the $\delta_0$ dimensional control vector of the own-ship; $\vec{u}_{j,\delta_j}(t)$ is the $\delta_j$ dimensional control vector of j-th met ship [22].

Control constraints and the state of the process are connected with the basic condition for the safe passing of the ships at a safe distance $D_s$ in compliance with the International Regulations for Preventing Collisions at Sea (COLREGs Rules)

$$
g_j(x_{j,\beta_j}, u_{j,\delta_j}) \leq 0
\tag{22}
$$

Goal function has the form of the payments, the integral payment and the final one

$$
I_{0,j} = \int_{t_0}^{t_k} [x_{0,\beta_0}(t)]^2 dt + r_j(t_k) + d(t_k) \ \rightarrow \ \min
\tag{23}
$$

The integral payment represents the additional distance traveled by the own-ship while passing the encountered ships and the final payment determines the final collision-risk $r_j(t_k)$ relative to the j ship and the final deflection of the own-ship $d(t_k)$ from the reference trajectory [23].

Two types of control goals were taken into consideration, programmed control $u_0(t)$ and positional control $u_0[x_0(t),t]$. The basis for the decision-making control are the decision-making patterns of the positional control processes, the patterns with the feedback arrangement representing the differential games.

While formulating the model of the control process, it is essential to take into consideration both the kinematics and the dynamics of the own-ship movement, the disturbances, the strategy of the encountered ships, and the assumed formula as the goal of the own-ship handling.

The diversity of selection of the possible models directly affects the synthesis of the own-ship control algorithms, which are afterwards affected by the ship-handling device, directly linked to the ARPA system and, consequently, determine the effects of safe and optimal control.

The application of reductions in the description of own-ship dynamics and the dynamics of the j-th encountered ship, and their movement kinematics, leads to the approximated model-positional and matrix.

### 4.2. Approximate Matrix Game Model

When leaving aside the own-ship dynamics equations, the general model of a differential game for the process of preventing collisions is reduced to the matrix game of J participants non-co-operating or co-operating among them. The state and control variables are represented by the values

$$
\begin{aligned}
&x_{j1} = D_j; \; x_{j2} = N_j; \; u_{01} = \psi; \; u_{02} = V; \; u_{j1} = \psi_j; \; u_{j2} = V_j \\
&j = 1,\, 2,\, ...,\, J
\end{aligned}
\tag{24}
$$

### 4.3. Matrix Game Anticollision (MGAC) Algorithm

Collision matrix risk R includes the values previously determined on the basis of data taken from anticollision system ARPA; the value of collision-risk $r_j$ with regard to the determined strategies of the own-ship and those of j-th encountered ships (Figure 7).

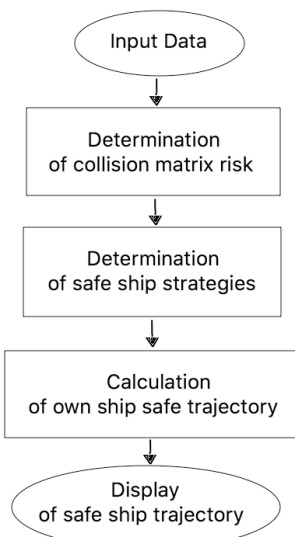

**Figure 7.** Matrix game anticollision (MGAC) algorithm block diagram.

The form of such a game is represented by the following risk matrix

$$
R = [r_j(\delta_0, \delta_j)]
\tag{25}
$$

containing the same number of columns as the number of participant O (own-ship) strategies. It has; e.g., a constant course and speed, course alteration 20° to starboard, 20° to port, etc., and contains a number of lines that correspond to a joint number of participants J (j-th encountered ships) strategies

$$
R = [r_j(\delta_0, \ \delta_j)] = \begin{vmatrix} r_{11} & r_{12} & \cdots & r_{1,\nu_0-1} & r_{1\nu_0} \\ r_{21} & r_{22} & \cdots & r_{2,\delta_0-1} & r_{2\delta_0} \\ \cdots & \cdots & \cdots & \cdots & \cdots \\ r_{\delta_1 1} & r_{\delta_1 2} & \cdots & r_{\delta_1,\delta_0-1} & r_{\delta_1\delta_0} \\ \cdots & \cdots & \cdots & \cdots & \cdots \\ r_{\delta_j 1} & r_{\delta_j 2} & \cdots & r_{\delta_j,\delta_0-1} & r_{\delta_j\delta_0} \\ \cdots & \cdots & \cdots & \cdots & \cdots \\ r_{\delta_J 1} & r_{\delta_J 2} & \cdots & r_{\delta_J,\delta_0-1} & r_{\delta_J\delta_0} \end{vmatrix}
\tag{26}
$$

The value of the collision-risk $r_j$ is defined as the reference of the current situation of the approach described by parameters $D^j_{min}$ and $T^j_{min}$ to the assumed assessment of the situation defined as safe and determined by the safe distance of approach $D_s$ and the safe time $T_s$—which are necessary to execute a maneuver avoiding a collision with consideration of actual distance $D_j$ between the own-ship and the encountered j-th ship (Figure 8)

$$
r_j = \left[ \varepsilon_1 \left( \frac{D^j_{min}}{D_s} \right)^2 + \varepsilon_2 \left( \frac{T^j_{min}}{T_s} \right)^2 + \varepsilon_3 \left( \frac{D_j}{D_s} \right)^2 \right]^{-\frac{1}{2}}
\tag{27}
$$

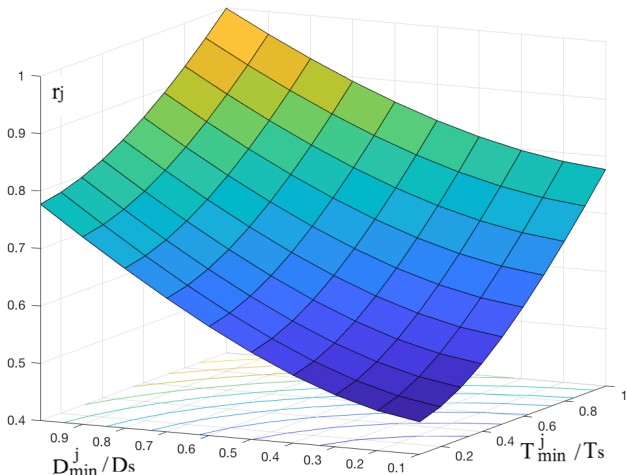

**Figure 8.** Example of the dependence of collision risk on the relative values of distance and time of approaching ships.

Weight coefficients $\varepsilon_1$, $\varepsilon_2$, $\varepsilon_3$ are dependent on the state of visibility at sea (good or restricted), the kind of water region (open or restricted), speed V of the ship, static L, and dynamic $L_d$ length of ship, and static B and dynamic $B_d$ beam of ship [10]

$$
\left. \begin{array}{l} 0.1 \leq \varepsilon_1 \leq 10 \\ 0.1 \leq \varepsilon_2 \leq 10 \\ 0.1 \leq \varepsilon_3 \leq 10 \end{array} \right\}
\tag{28}
$$

$$
L_d = 1.1 \ (1 + 0.345 \ V^{1.6})
\tag{29}
$$

$$
B_d = 1.1 \ (B + 0.767 \ LV^{0.4})
\tag{30}
$$

Assuming higher values of particular weight coefficients $\varepsilon_1$, $\varepsilon_2$ and $\varepsilon_3$; the share of the risk of collision $r_j$ depending on the distance of ships $D_j$ and $D^j_{min}$ or the time of excessive approach of ships $T^j_{min}$ increases.

The constraints affecting the choice of strategies are a result of the recommendations of the way of priority at sea. Player O (own-ship) may use $\delta_0$ of various pure strategies in a matrix game and player J (encountered ships) has $\delta_j$ of various pure strategies.

As the game, most frequently does not have a saddle point, the state of balance is not guaranteed, and there is a lack of pure strategies for both players in the game. In order to solve this problem, dual linear programming may be used.

In a dual problem, player O, having $\delta_0$ various strategies to be chosen, tries to minimize the risk of collision

$$I_0 = \min_{\delta_0} r_j \tag{31}$$

while player J, having $\delta_j$ strategies to be chosen, tries to maximize the collision-risk

$$I^j = \max_{\delta_j} r_j \tag{32}$$

For a non-co-operative matrix game, the problem of determining an optimal strategy may be reduced to the task of solving a dual linear programming problem [24]

$$\left(I_0^j\right)^* = \min_{\delta_0} \max_{\delta_j} r_j \tag{33}$$

For a co-operative matrix game, the problem of determining an optimal strategy may be reduced to the task of solving a dual linear programming problem

$$\left(I_0^j\right)^* = \min_{\delta_0} \min_{\delta_j} r_j \tag{34}$$

Mixed strategy components express probability distribution $P = [p_j(\delta_0,\delta_j)]$ of players using pure strategies

$$P = [p_j(\delta_0,\ \delta_j)] = \begin{vmatrix} P_{11} & P_{12} & \cdots & P_{1,\delta_0-1} & P_{1\delta_0} \\ P_{21} & P_{22} & \cdots & P_{2,\delta_0-1} & P_{2\delta_0} \\ \cdots & \cdots & \cdots & \cdots & \cdots \\ P_{\delta_1 1} & P_{\delta_1 2} & \cdots & P_{\delta_1,\delta_0-1} & P_{\delta_1\delta_0} \\ \cdots & \cdots & \cdots & \cdots & \cdots \\ P_{\delta_j 1} & P_{\delta_j 2} & \cdots & P_{\delta_j,\delta_0-1} & P_{\delta_j\delta_0} \\ \cdots & \cdots & \cdots & \cdots & \cdots \\ P_{\delta_J 1} & P_{\delta_J 2} & \cdots & P_{\delta_J,\delta_0-1} & P_{\delta_J\delta_0} \end{vmatrix} \tag{35}$$

The solution for the steering goal is the strategy of the highest probability and will also be the optimal value approximated to the pure strategy

$$\left(u_0^{\delta_0}\right)^{\bullet} = u_0^{\delta_0}\left\{\left[p_j\left(\delta_0,\delta_j\right)\right]_{max}\right\} \tag{36}$$

The safe trajectory of the own-ship was treated here as a sequence of changes to the course and speed. The established values are as follows: safe passing distances among the ships under given visibility conditions at sea $D_s$, time delay of manoeuvring and the duration of one stage of the trajectory as one calculation step. At each step, the most dangerous ship is determined with regard to the value of collision risk $r_j$.

Consequently, on the basis of the semantic interpretation of the COLREGs, the direction of a turn of the own-ship is selected to the most dangerous encountered ship [25–27].

Collision matrix risk R is determined for the admissible strategies of the own-ship $\delta_0$ and those $\delta_j$ for j-th ships encountered. By applying dual linear programming in order to solve the matrix game, we obtain the optimal values of the own-ship course and those of the j-th ship at the smallest deviation from their initial values.

If, at a given step, no solution can be found at the speed of the own-ship V, the calculations are repeated at reduced speed by 25% until the game is solved.

The calculations are repeated step by step until the moment when all elements of matrix R become equal to zero, and the own-ship, after having passed the encountered ships, returns to its initial course and speed.

Using the *linprog* function, that is, linear programming from the *optimtool* optimization toolbox contained in the MATLAB_R2017a software, the matrix game anti-collision (MGAC) computer program was designed for the determination of the safe own-ship trajectory in a collision situation [10].

## 5. Research Results

The aim of the computer simulation research of the FNAC and MGAC algorithms to determine the optimal safe-ship trajectory in collision situations was to evaluate methods to solve the problem formulated in this work by using fuzzy-set theory as a multistage and matrix game theory as a multistep decision-making process.

The computer simulation of the FNAC and MGAC algorithms was carried out in MATLAB software on an example of the real navigational situation of passing J = 3 encountered ships in the Skagerrak Strait in good visibility $D_s = 0.2$ nm and the restricted visibility $D_s = 2.0$ nm (nautical miles) (Table 1).

**Table 1.** Data of own-ship and met ships: 1, 2 and 3.

|  | Bearing $N_j$ (°) | Distance $D_j$ (nm) | Speed $V_j$ (kn) | Course $\psi_j$ (°) |
|---|---|---|---|---|
| Own-ship | - | - | 20 | 0 |
| Ship 1 | 326 | 8.8 | 13.5 | 90 |
| Ship 2 | 6 | 14.3 | 16.2 | 180 |
| Ship 3 | 11 | 7.5 | 16.0 | 200 |

The situation was registered on board r/v HORYZONT II, a research and training vessel of the Gdynia Maritime University, on the radar screen of the ARPA anticollision system Raytheon. The sample results of the performed computer simulations for the navigational situation when the own-ship passed three met ships in a good and restricted visibility at sea are presented in Figures 9 and 10, respectively.

The trajectories of the own-ship, shown in Figures 9 and 10, are optimal, but in different ways.

The FNAC trajectory ensures minimum risk of collision of own-ship, taking into account the uncertainty of the control process, described by the fuzzy set membership functions of state and control constraints, and collision risk, not including maneuvers of encountered ships.

On the other hand, the trajectory of MGAC ensures a minimum collision-risk of the own-ship taking into account the maneuvering of encountered ships in a co-operative or non-co-operative way.

Designated safe trajectories FNAC and MGAC of the own-ship are the reference trajectories for automatic ship's control systems using the autopilot and main engine.

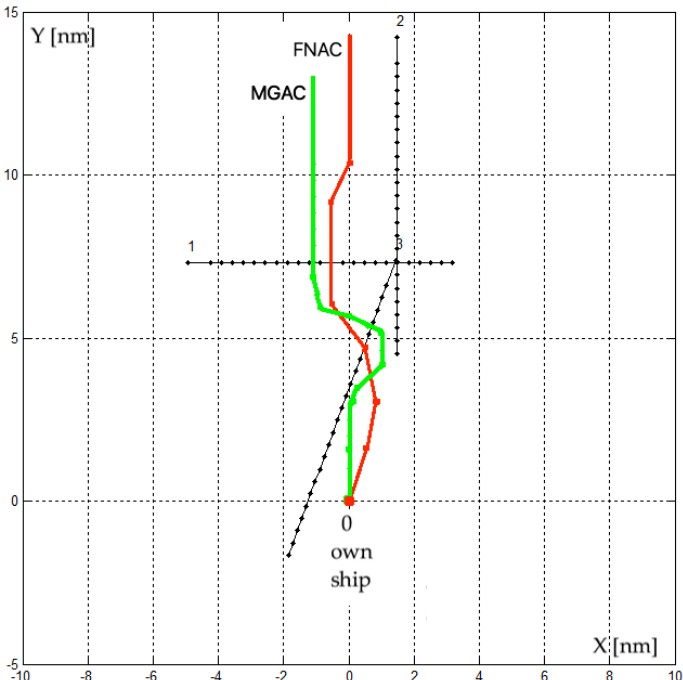

**Figure 9.** Comparison of safe and optimal trajectories of own-ship in the situation of passing three ships, in conditions of good visibility at sea at $D_s$ = 0.2 nm, determined using the following algorithms: FNAC decision chosen trajectory = 0.5766; MGAC final payment = 1.08 nm.

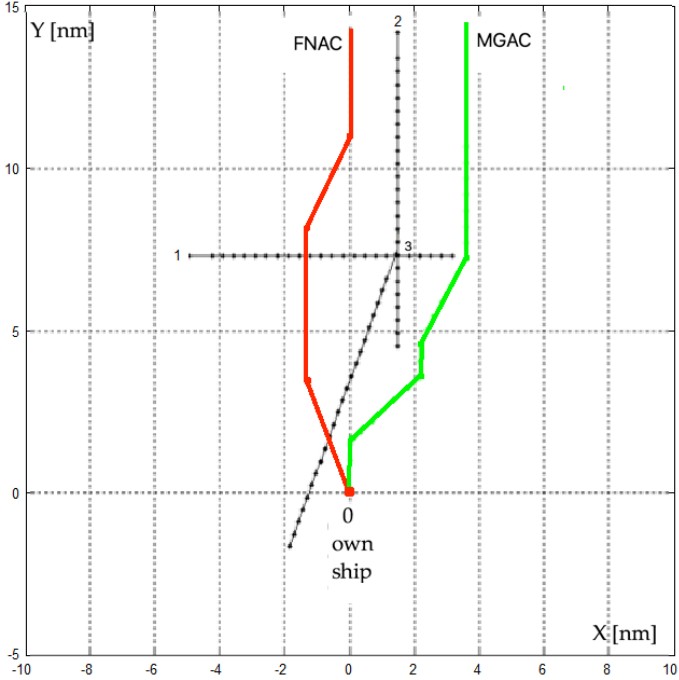

**Figure 10.** Comparison of safe and optimal trajectories of own-ship in the situation of passing three ships, in conditions of restricted visibility at sea at $D_s$ = 2.0 nm, determined using the following algorithms: FNAC decision chosen trajectory = 0.6452; MGAC final payment = 3.83 nm.

## 6. Conclusions

This work showed that the proposed concept of applying the fuzzy neural anticollision and matrix game anticollision algorithms is a promising way to solve the considered problem and design a novel anticollision system, allowing to increase the safety of maritime transport.

The FNAC algorithm achieves a short computation time of about one second, while the MGAC algorithm scope of the calculation time is from about two to five seconds.

The obtained solutions obtained using both algorithms meet the requirements of the COLREGs.

The MGAC computer program, designed in MATLAB software, takes into consideration the following: degree of co-operation with the own-ship and encountered ships, COLREGs Rules, advance time for a maneuver calculated with regard to the own-ship dynamic features, and the assessment of the final deviation between the real and reference trajectories.

In summary of these results, the created algorithms can be used as a tool to assist navigator in making maneuver decision, in complex collision situations, when passing through more ships, especially in restricted visibility at sea.

In future works, the sensitivity analysis of safe ship control should be performed to change the parameters of the process model and the inaccuracy of information from the ARPA radar system; moreover, the design of the ARPA system may be considered, extended with the function of a computer-aided maneuvering navigator decision, using the FNAC and MGAC algorithms.

**Author Contributions:** Conceptualization, J.L.; Methodology, M.M.-S.; Software, M.M.-S.; Validation, J.L. and M.M.-S.; Formal analysis, J.L. and M.M.-S.; Investigation, J.L. and M.M.-S.; Resources, J.L. and M.M.-S.; Data curation, M.M.-S.; Writing—original draft preparation, M.M.-S.; Writing—review and editing, J.L.; Visualization, J.L.; Supervision, J.L.; Project administration, J.L.; Funding acquisition, M.M.-S.

**Funding:** This research was funded by a statute research project of Gdynia Maritime University in Poland, No. 446/DS/2018: "Design and simulation tests of marine automation systems in MATLAB/Simulink and LabVIEW software".

**Conflicts of Interest:** The authors declare no conflict of interest regarding the publication of this paper. The funders had no role in the design of the study; in the collection, analyses, or interpretation of data; in the writing of the manuscript; or in the decision to publish the results.

## Nomenclature

| | |
|---|---|
| $A_p$ | axonic activation |
| C | fuzzy-set goal |
| D | fuzzy-set decision |
| $D_s$ | safe distance of approach |
| $D_j$ | distance between own-ship and the j-th met ship |
| DCPA | distance to closest point of approach |
| G | fuzzy-set contraints |
| P | probability distribution |
| R | collision-risk matrix |
| $r_j$ | value of the collision-risk |
| $u_t$ | controls |
| TCPA | time to closest point of approach |
| $T_s$ | safe time of approach |
| U | control-set |
| $u_k(t)$ | postsynaptic activation at stage t |
| V | ship speed |
| $V_{opt}$ | optimal ship speed |
| W | set of final states |
| $X_{t+1}, X_t$ | ship position co-ordinates |

| | |
|---|---|
| X | set of real ship position co-ordinates |
| $\alpha_t$ | axonic threshold at stage t |
| $\mu_R$ | membership function of fuzzy-set collision-risk |
| $\mu_{Rsafe}$ | value of $\mu_R$ at which the process is assumed safe |
| $\lambda_c, \lambda_d, \lambda_{rd}, \lambda_{rt}$ | navigator's subjective parameters |
| $\psi$ | ship course |
| $\psi_{opt}$ | optimal ship course |
| $\wedge$ | minimum operator |

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
