# Peer review of "Comparison of Computational Intelligence Methods Based on Fuzzy Sets and Game Theory in the Synthesis of Safe Ship Control Based on Information from a Radar ARPA System"

_remotesensing, doi:10.3390/rs11010082_

Round 1

Reviewer 1 Report

Article is interesting, the subject is current and has useful value. The obtained results are interesting. In my opinion, the article lacks a formal comparison of both methods. In addition, in the case of the FNAC method, the significance of the obtained results, which are included in the description of figures 8 and 9 as the decision chosen trajectory, is not explained. The above comments do not affect my high appreciation of the substantive content of the article.

Author Response

Thank you for the valuable comments of the Reviewer, which contributed to improving quality of article.

The article was edited by MDPI’s English editing service.

All remarks of the Reviewer have been included in the text of the revised article. 

Please find attached an amended article with redesigns marked in red color.

Reviewer 2 Report

This paper proposes several approaches in solving ship collision avoidance problem. FNAC and MGAC were applied in optimal safe ship trajectory. However several points should be more clear and proven in academic way.

1.Equations or all Nomenclature in this article should be clearly explained. Most of them are not described in detail such as equation (2), (4), (6), --- (22), (23), (36) and so on. Therefore it is very hard to clearly understand the purpose and the process of this article.

2. The FNAC is very crucial research approach in this article. However followings are not described enough to give clear understanding.  

(1) The structure of NN and data(input and output) process used for the NN

(2) How the NN actually are used for safe ship trajectory?   IN figure 2, NN was not mentioned, where is the control concept figure which include NN?

3. In fig 8 and 9, more evaluation or comments on the simulation results should be described.

4. In fig 8 and 9, they show the simulation results only with trajectories. Other data (variables; speed, rudder angle, heading angle) should be showed in time history simultaneously

5. In fig 8 and 9, own ship have two trajectories according to the FNAC and MGAC. Which one is optimal ship trajectory?

 6. General concept for suggested safe ship control optimization design should be depicted in one figure. It should include input and output data including suggested control method such as FNAC and game algorithm. In this article Fig 2 and Fig describe control concept, however it is not enough to understand the structure of control algorithm. 

 General review

 This article tried new approaches in ship collision avoidance problem and apparently showed simulation results. However more detailed explanation on how suggested algorithms were applied in the system is required. Also Equations or all Nomenclature used in article should be depicted to be published technical journal.

Author Response

(The authors gave the same response as above.)

Reviewer 3 Report

I am confident that the topic of the article is interesting and essential in terms of research. The results are important and useful for the practice. In order to enhance the article quality, I suggest the following remarks be taken into account:

1.       Figure 1: Please provide the meaning of notations in figure.

2.       Line 35: Please delete ‘most’ (seems to be a little too strong a statement).

3.       The literature review should be extended to include the latest publications on intelligent solutions in shipping that refer to the discussed subject, for instance:

·       „Inference engine in an intelligent ship course-keeping system” Computational Intelligence and Neuroscience vol. 2017, art. no. 2561383, 2017 (1-9)

·     

4.       Equation 2: Please add ‘=’.

5.       Equation 3: Incorrect expression.

6.       Equation 4: Please provide the meaning of notations in equation (lambda).

7.       Equation 28: Incorrect expression.

8.       Figures 7-9: The authors should add units.

9.       Figures 8-9 seems to be insufficiently described.

10.     It is good idea to add 1-3 sentences about future analysis.

Author Response

(The authors gave the same response as above.)

Reviewer 4 Report

Dear authors,

I found that your research paper develops a  very interesting topic of high scientific soundness, very attractive for a large community of readers.

I feel that the overall quality of your research article can be improved based on the following comments:

Please extend the literature review from Section 1 with the more recent contributions in the field, especially from the last two years, 2017 and 2018. Update, according to these additions, the Section References with new items.  

To eliminate any possible  notation  confusion for matrix R, I feel that is better to write in the line 211  of the manuscript " The collision matrix risk R" instead of "the matrix game R. 

Please could you  make an impact study on different settings of the weight coefficients ε1,ε2,ε3  constrained to the values given by (28) on the evolution of the risk of collision rs factors given in (27)? What is reflected this impact on the overall performance of the  proposed MGAC algorithm?

I consider that to better delineate where the first proposed FNAC algorithm  begins and ends, it is necessary to  introduce a new  separate section dedicated to develop this algorithm that includes the sections 2.1-2.3 and entire section 3. 

For consistency, could you formulate the second proposed algorithm MGAC in a similar way as FNAC, showing the steps or at least one new detailed  flowchart,  to be followed by the interested readers in the field  to implement it?

For Land Bgiven in (29) and (30) please precise also the source of their definition.

Please provide some details on the use of subroutine MATLAB's Optimisation Toolbox  function "linprog", and mention the MATLAB version used. 

Precise the software package and the version used to implement FNAC algorithm. 

Precise if you use the SIMULINK simulations, and ii is the case, show in a separate figures the SIMULINK models developed to implement MGAC algorithm. 

I expect also a rigorous analysis of overall performance of both proposed algorithms in Section 6, and some details on   their robustness to changes in the parameters values, especially in the weight coefficients mentioned in the third comment.

Thanks,

Author Response

(The authors gave the same response as above.)

Reviewer 5 Report

Figures  do not carry scientific components. Enough simplified representation.

not enough reference on neural network technology. The article is focused on a more factor analysis. The vagueness of the system is not pronounced. .

Author Response

(The authors gave the same response as above.)

Round 2

Reviewer 2 Report

Most of comments were reflected in the revised version.

Reviewer 3 Report

The authors addressed my previous comments. I am satisfied with this revision.

Reviewer 4 Report

Dear Authors,

Thank you very much for all your efforts to improve the overall quality of this  manuscript.

Reviewer 5 Report

The main directions were taken into account. Work looks better.